# Controlled Release in Hydrogels Using DNA Nanotechnology

**DOI:** 10.3390/biomedicines10020213

**Published:** 2022-01-19

**Authors:** Chih-Hsiang Hu, Remi Veneziano

**Affiliations:** Department of Bioengineering, College of Engineering and Computing, George Mason University, Manassas, VA 20110, USA; chu6@gmu.edu

**Keywords:** hydrogel, gelatin, DNA nanotechnology, strand displacement, temporal release

## Abstract

Gelatin is a biopolymer widely used to synthesize hydrogels for biomedical applications, such as tissue engineering and bioinks for 3D bioprinting. However, as with other biopolymer-based hydrogels, gelatin-hydrogels do not allow precise temporal control of the biomolecule distribution to mimic biological signals involved in biological mechanisms. Leveraging DNA nanotechnology tools to develop a responsive controlled release system via strand displacement has demonstrated the ability to encode logic process, which would enable a more sophisticated design for controlled release. However, this unique and dynamic system has not yet been incorporated within any hydrogels to create a complete release circuit mechanism that closely resembles the sequential distribution of biomolecules observed in the native environment. Here, we designed and synthesized versatile multi-arm DNA motifs that can be easily conjugated within a gelatin hydrogel via click chemistry to incorporate a strand displacement circuit. After validating the incorporation and showing the increased stability of DNA motifs against degradation once embedded in the hydrogel, we demonstrated the ability of our system to release multiple model cargos with temporal specificity by the addition of the trigger strands specific to each cargo. Additionally, we were able to modulate the rate and quantity of cargo release by tuning the sequence of the trigger strands.

## 1. Introduction

Hydrogels are three-dimensional (3D) hydrophilic networks of crosslinked polymers that can be designed to mimic the structural, biochemical, and mechanical properties of the extracellular matrix (ECM). Hydrogels function as cellular support and promote the distribution of biochemical factors between cells [1,2]. Hydrogel constructs are used in several biomedical applications, such as regenerative medicine [3,4,5,6,7], delivery systems [8,9,10], and as bioink for bioprinting [11,12]. Most of the hydrogels commonly used can be organized into three different categories: synthetic, natural, or composite, which includes both synthetic and natural components [13,14,15].

Synthetic polymers, such as poly(N-isopropylacrylamide) (PNIPAM), poly(ethylene glycol) (PEG), and poly(acrylamide) (PAAM), have the advantage of being chemically homogeneous, easy to manufacture, with tunable physicochemical properties, and a controlled degradation rate, compared to natural polymers [13,15,16,17,18]. However, synthetic polymers tend to lack the biological cues that can be found in some natural polymers, such as the RGD peptide found in collagen, which enable cell adhesion [19,20]. 

Thus, these synthetic polymers require additional synthesis steps to incorporate biological cues within the hydrogel, which can limit their biological applications [13,18]. In addition, natural polymers, such as gelatin, collagen, and fibronectin, have higher biocompatibility, biodegradability, and better mimic the native ECM [3,20,21,22,23,24]. The biological benefits of natural polymers have sparked research interest in utilizing them for smart drug delivery systems [8,9,25], tissue engineering applications [3,7,26], and 3D bioprinting [11,27,28,29], despite the drawback of uncontrolled degradation, limited mechanical properties, and batch-to-batch variation compared to the more controlled formulation of synthetic hydrogels [13,18,30].

Hydrogels can be easily loaded or modified with biomolecules (i.e., cytokines and growth factors) for controlling cellular behavior. However, despite the ease of incorporation of biochemical signals, those systems do not allow for the precise temporal release of biomolecules to mimic the natural sequential releases of biological factors, which is one of the major challenges in regenerative medicine and drug delivery systems. A potential solution to this problem is to leverage DNA nanotechnology tools to build a DNA-based circuit for the temporal release of biomolecules, and this can be integrated in existing hydrogels [31,32]. 

Indeed, DNA motifs can be designed to any shape and size and easily modified with multiple biomolecules that can be released on-demand with a strand displacement reaction (SDR) to achieve the temporal controlled release of biomolecules. SDR has already been used to design DNA circuits [33,34,35], drug release systems [36,37,38], and to actuate the mechanical properties and control cell adhesion on a surface-based assay [39]. 

The basic principle of SDR relies on the use of a single strand DNA, called a trigger strand, that can bind to a DNA complex and displace a strand from the complex [40,41,42]. The displacement reaction occurs because the trigger strand has a longer hybridization region, known as the toehold region, compared to the initial strand. The displacement kinetics can be tuned by changing the length of the toehold region, thus, making the displacement reaction faster or slower [40,43]. 

Additionally, DNA nanostructure design and sequence flexibility allow for designing motifs that facilitate the attachment of various cargo simultaneously with precise control of their stoichiometry to enable multiplexed release. The ease of DNA functionalization allows the utilization of various chemistry to attach biomolecules. However, despite its potential, the combination of DNA nanostructures, such as multi-way junctions carrying SDR motifs to create a temporal release circuit that can deliver multiple cargos with temporal specificity has not yet been demonstrated in hydrogels. 

In this study, we synthesized DNA motifs that include a SDR system and that can be easily linked to gelatin hydrogel or potential other biopolymer hydrogels, such as collagen hydrogels, via robust click chemistry, and we demonstrated the ability of this system to release specific cargo with temporal specificity by the addition of the corresponding trigger strand for each cargo as illustrated in Figure 1. Additionally, we were able to demonstrate the capability to tune the quantity and the rate of cargo release with our SDR system by modifying our trigger strand sequences with the introduction of precisely located mismatches. 

This further expanded the versatility of the system we developed. It is important to note that the stability of the DNA motif in physiological conditions was greatly improved upon incorporation into the gelatin hydrogels, which will be useful for cellular assays. Overall, our SDR-controlled release hydrogel system represents a promising approach for controlling the release of cargo for a better understanding of the temporal effect of specific biomolecule on cellular behavior. 

## 2. Materials and Methods

### 2.1. Materials

Bovine skin gelatin (Type A, G1890), Dibenzocyclooctyne-PEG4-N-hydroxysuccinimidyl ester (DBCO-PEG4-NHS ester, 764019), N-hydroxysuccinimidyl ester (NHS, 1s30672), fluorescamine (F9015), and 30 kDa (UFC5030) and 100 kDa (UFC5100) Amicon^®^ Ultra Centrifugal Filters were purchased from Sigma-Aldrich (St. Louis, MO, USA). All oligonucleotides (Appendix A) were purchased from Integrated DNA Technologies (Coralville, IA, USA). 2,4,6-Trinitrobenzene Sulfonic Acid (TNBSA, 28997) assay, 10X PBS (J75889), and 7 kDa and 40 kDa Zeba^TM^ Desalting Columns were purchased from Thermo Scientific (Waltham, MA, USA). 

N-(3-dimethylaminopropyl)-N’-ethylcarbodiimide (EDC HCl, AS-29855) was purchased from AnaSpec (Fremont, CA, USA). Dimethylformamide (DMF, 97064-586) was purchased VWR (Radnor, PA, USA). Water, molecular biology grade (351-029-101) was purchased from Quality Biological (Gaithersburg, MD, USA). Bullseye General Purpose Agarose (BE-A125) was purchased from MidSci (St. Louis, MO, USA). QuickLoad 100 bp DNA ladder was purchased from New England BioLabs^®^ (Ipswich, MA, USA).

### 2.2. Gelatin Reactive Amine Group Quantification

We used the TNBSA assay to determine the quantity of reactive amine group available on gelatin prior to performing the DBCO-PEG4-NHS modification. The TNBSA assay protocol was adapted from on the manufacture’s instruction. Briefly, gelatin solution in 0.1 M sodium bicarbonate (pH 8.5) was added with TNBSA and incubated for 2 h before adding 10% SDS and 1 N HCl. The absorbance was measured at a wavelength of 335 nm. A glycine solution with a known concentration was used to prepare the standard curve following the same protocol.

### 2.3. Gelatin Modification with DBCO Groups

#### 2.3.1. DBCO Modification of Gelatin

Gelatin was modified with DBCO-PEG4-NHS ester in 1× PBS (pH 8.3). The degree of modification was modified by adding varying amounts of DBCO-PEG4-NHS ester. DBCO-PEG4-NHS ester stock was prepared in DMF at 10 mg/mL (15.39 mM). The reaction was carried out overnight at 37 °C in 1× PBS (pH 8.3) with a final gelatin concentration of 12.5 mg/mL in the reaction solution. 

Unreacted DBCO-PEG4-NHS ester was filtered out using 7 kDa Zeba^TM^ Spin Desalting Columns according to the manufacturer’s instruction with 1× PBS (pH 8.3) used for the washing step. The solution was labeled as ‘xx’% DBCO-Gelatin (DG) and stored at 4 °C for further characterization and modification. The ‘xx’ in the labeling stands for the targeted percent modification.

#### 2.3.2. Conjugation of the Azide-DNA Bait to the DBCO-Gelatin (DG)

DG was functionalized with azide-DNA by strain promoted alkyne-azide cycloaddition (SPAAC). The degree of modification was tuned by adding varying the amount of azide-DNA strands. The reaction was carried out overnight at room temperature in 1× PBS (pH 8.3) with a final gelatin concentration of 1 mg/mL in the reaction solution. Unreacted azide-DNA was filtered out with 30 kDa Amicon^®^ Ultra Centrifugal Filters at 4000× *g* with three washes using 1× PBS (pH 8.3) as the washing buffer. The solution was labeled as ‘xx’% DNA-DG and stored at 4 °C. The ‘xx’ in the labeling stands for the targeted percent modification.

### 2.4. DBCO Linker and DNA Bait Modification Quantification

#### 2.4.1. Quantification of the Gelatin Modification with DBCO

The degree of DBCO modification was determined using fluorescamine. Glycine Standard with a concentration ranging from 0 to 500 µM was prepared to quantify the remaining free amine group after conjugation with DBCO. The unmodified and modified gelatin samples were diluted down to 0.75 mg/mL (90 µL). Fluorescamine solution was prepared by dissolving it with acetone immediately before its use at a concentration of 3 mg/mL. 

Fluorescamine was added to each well (30 µL). The fluorescence intensity of fluorescamine was recorded with a Tecan Safire^2^ fluorescence microplate reader. Fluorescamine was excited at a wavelength of 380 nm, and the emission spectrum was recorded from 450 to 750 nm with step size of 2 nm. The plate was shaken for 10 s and allowed to settle for 30 s prior to each measurement. The degree of substitution was determined by calculating the reactive amino group difference between the modified and unmodified gelatin from the fluorescamine assay.

#### 2.4.2. Quantification of the Efficiency of DNA Bait Conjugation

The degree of DNA overhang conjugation to gelatin polymer was quantified by hybridizing a complementary DNA strand modified with a HEX fluorophore. This strand is called the DNA probe strand and is used in a 5× molar excess to the estimated amount of DNA overhang conjugated on gelatin. The amount of DNA probe strand used was determined by assuming 100% reaction efficiency for the azide-DBCO reaction and the DBCO amount was obtained from the result of the fluorescamine assay. 

The mixture (50 µL) was incubated in a Bio-Rad T100^TM^ Thermal Cycler from 37 °C to 4 °C over 25 min, and excess DNA probe strands were removed using filtration with 30 kDa Amicon^®^ Ultra Centrifugal Filters using three washes with fresh 1× PBS (pH 8.3). The fluorescence intensity of HEX was recorded with a Tecan Safire^2^ fluorescence microplate reader with an excitation wavelength of 495 nm, and the emission spectrum was recorded from 535 to 700 nm (step size: 2 nm). A standard curve of the DNA probe strand was used to calculate the amount of DNA probe strand attached. 

### 2.5. DNA Motifs Folding

The DNA motifs were folded by mixing an equimolar concentration of all the DNA strands at 5 µM in 1× folding buffer (40 mM Tris-base, 20 mM acetic acid, 2 mM EDTA, 12 mM MgCl_2_, and pH 8.0) and incubated in a Bio-Rad T100^TM^ Thermal Cycler. The cycles were set to have the temperature decreasing from 95 to 4 °C over the course of 2 h.

### 2.6. DNA Motifs Characterization

#### 2.6.1. Agarose Gel Electrophoresis

Agarose gel electrophoresis (1.5% agarose in Tris Acetate EDTA (TAE) buffer) was used to determine the correct folding of both 3-way junction (3WJ) and 4-way junction (4WJ). A volume of 30 µL of DNA nanostructure sample prepared at 750 nM in 1× folding buffer was loaded in a gel that was run at 110 V for 30 min. The QuickLoad 100 bp DNA ladder from New England BioLabs^®^ (Ipswich, MA, USA) was used along with the samples. The agarose gel was imaged with the Azure C150 imager.

#### 2.6.2. qPCR Melting Curve

The Q qPCR machine from Quantabio was used to obtain the melting curve for both 3WJ and 4WJ. Bare 3WJ and 4WJ was diluted down to 750 nM in 1× folding buffer with 1× SYBR Green I. The qPCR program was set to have an activation step at 40.0 °C for 5 min followed by two cycles at 40.0 °C with cycling time of 60 s. The temperature range for the melting step was set to be from 40.0 to 95.0 °C with step size of 0.5 °C/s. 

The melting curve was derived from the fluorescence of SYBR Green by taking the first derivative of the fluorescence response. This step was done automatically by the Q qPCR software (V1.0.1, QuantaBio, Beverly, MA, USA). 

#### 2.6.3. Fluorescence Measurements

Three fluorophores and two quenchers were used for the study. The three fluorophores were HEX, FAM, and Texas-Red. The two types of quenchers were Iowa Black FQ and Iowa Black RQ. Both HEX and FAM fluorophores were paired with Iowa Black FQ, and Texas-Red was paired with Iowa Black RQ. The fluorescence measurement for each combination was done with a fluorescent microplate reader with a specific excitation and emission dependent on the fluorophore used. The excitation wavelength of HEX was 495 nm, and the emission spectrum was recorded from 535 to 800 nm. The excitation wavelength of FAM was 455 nm, and the emission spectrum was recorded from 500 to 800 nm. The excitation wavelength of Texas-Red was 555 nm, and the emission spectrum was recorded from 590 to 800 nm. 

#### 2.6.4. Dynamic Light Scattering (DLS) Size Measurement

DNA motifs, both 3WJ and 4WJ were diluted to 2 µM with water and 60 µL of the solution was used for the measurement to determine their hydrodynamic diameter and their distribution. The Malvern Nano-ZS was used for the measurement. The measurement angle was 173° backscatter. 

#### 2.6.5. Stability Study in Serum

DNA-Gel hydrogel was formed based on the method mentioned below in Section 2.8 with either 4WJ-HEX with quencher or 4WJ-HEX without quencher, and 70 µL of gel was cast into a black 96-well plate. We added 20% serum on top of the gel followed by addition of 30 µL of mineral oil to minimize evaporation. We added 15 µL of DNAse (2000 U/mL) or 1× PBS (pH 8.3) on top of the control gels followed by 30 µL of mineral oil to minimize evaporation. The DNAse treatment was also used for 4WJ-HEX in solution. The fluorescence intensity of HEX was measured over the course of 24 h. 

### 2.7. Synthesis of the 4WJ-Gelatin (DNA-Gel) with DG and DNA Motifs

DNA-Gel was synthesized by mixing the DNA bait modified gelatin solution (0.1% *w*/*v* gelatin) and DNA motifs (5 µM) in 1× PBS (pH 8.3). The mixture (50 µL) was incubated in a Bio-Rad T100^TM^ Thermal Cycler from 37 to 4 °C over the course of 25 min. The final concentration of the gelatin in the mixture was 0.01% *w*/*v* and the concentration of DNA motif is 2.5 µM. Buffer exchange was carried out after incubation with 30 kDa Amicon^®^ Ultra Centrifugal Filters using three washes with fresh 1× PBS (pH 7.4). DNA-Gel is stored in 4 °C until further usage.

### 2.8. Gelatin and DNA-Gel Hydrogel Formation

Gelatin hydrogels are crosslinked via EDC/NHS chemistry. The molecular weight of the gelatin is assumed to be 100 kDa, and the carboxyl group concentration was assumed to be 0.77 mmol.g^−1^ based on previous literature [44]. The EDC was used at a 5× molar excess to the carboxyl group presented on gelatin, and NHS was used at an equimolar ratio to the EDC. The final gelatin concentration in the gel was 5% *w*/*v*. The crosslink process was carried out in 1× PBS (pH 7.4) at room temperature, and the gel was allowed to polymerize for one hour. The crosslink process was carried out immediately prior to the usage of the hydrogel. DNA-Gel hydrogels were synthesized following the same procedure as the pure gelatin hydrogel with additional DNA-Gel incorporated in the gelation process. Detailed volumes and the initial concentrations of each component are shown in Table 1. Basic characterization was carried out, such as a swelling study [44,45] and qualitative gel imaging, at room temperature. 

### 2.9. Controlled Release Study

#### 2.9.1. Controlled Release in Solution

The ‘Q’ qPCR machine from Quantabio was used to quantify the release of quencher from the DNA motifs by measuring the increase in fluorescence intensity for each fluorophore at 40 °C. The ‘Q’ qPCR program was set to run for 60 cycles with 20 s of cycling time, and the cycling time increased for 5 s for each cycle, i.e., 25 s of cycling time for cycle two and 30 s for cycle three. The total run time for the procedure was about 2 h and 50 min. The excitation for HEX was 540 nm, and the emission was recorded at 570 nm. The excitation for FAM was 465 nm, and the emission was recorded at 510 nm. The excitation for TR was 585 nm, and the emission was recorded at 618 nm. The release was normalized against the positive control, which was the DNA motif without the quencher. 

#### 2.9.2. Controlled Temporal Release in DNA-Gel Hydrogel

The DNA-Gel hydrogel is synthetized with the procedure shown above in Section 2.7 with a final concentration of 4WJ fixed as 500 nM. A DNA-Gel hydrogel without the quencher was used as the positive control, and a DNA-Gel hydrogel without the addition of trigger solution was used as the negative control. The trigger solution was diluted to match the 1:1 ratio between the trigger strand and 4WJ in the gel based on the estimated 4WJ within DNA-Gel. 

We added 15 µL of the diluted trigger on top of the DNA-Gel hydrogel, and 15 µL of 1× PBS (pH 8.3) on top of the control groups. We added 30 µL of mineral oil on the top of each sample to minimize the evaporation during measurement. The fluorophores emission spectra were measured with a Tecan Safire^2^ fluorescence microplate reader over 17 h. The release was normalized against the positive control, which was the DNA motif without the quencher. 

### 2.10. Statistical Analysis

All the values are reported as the mean ± standard deviation. A Welch two sample t-test was used to determine the significance between two variables. The Kruskal–Wallis rank sum test and Dunn’s Kruskal–Wallis multiple comparison were used as post-hoc analysis to analyze the final release quantity at the release study end point. All statistical analyses were conducted with RStudio [46]. 

## 3. Results and Discussion

### 3.1. Design and Synthesis of the DNA Motifs for the Strand Displacement Release Circuit 

#### 3.1.1. Design of DNA Motifs

To enable control over the sequential release of multiple biological signals and to reduce the need for multiple conjugation steps, we used DNA-based assemblies. Specifically, we designed DNA motifs, namely a three-way (3WJ) and a four-way junction (4WJ), as shown in Figure 2a, that can simultaneously incorporate a multi SDR system and multiple cargos on a single motif (two for the 3WJ and three for the 4WJ). To facilitate the tracking of release and demonstrate the feasibility of our method, we used fluorophores (HEX, FAM, and Texas-Red) and their respective quenchers (Iowa Black RQ or FQ) as model cargos in this study. 

These structures were chosen because of their relative simplicity, stability, ease of synthesis, and conjugation. Additionally, these DNA motifs have been used in prior studies as building blocks to create sensor and/or hydrogel complexes increasing the potential of our strategy [47,48]. We designed orthogonal overhangs to be displayed by the DNA motifs to facilitate the specific attachment of different cargos. For the strand displacement circuit, we implemented sequences developed by Scalise et al. with the addition of linker sequences to allow incorporation onto our nanostructures [41]. The full list of DNA sequences designed for the DNA motifs and strand displacement circuit can be found in the Appendix A.

#### 3.1.2. Assembly and Folding Characterization

The DNA motifs were folded with a 2-h annealing protocol and agarose gel electrophoresis was carried out for both 3WJ and 4WJ to assess proper folding of the DNA nanostructures. Figure 2b shows that both 3WJ and 4WJ were properly assembled with no by-products as expected for this type of structure assembled with an equimolar concentration of strands. As shown in Figure 2b, the size indicated on the agarose gel is comparable to the expected size for both 3WJ and 4WJ, which are 207 bases (b) and 276 b, respectively. 

To determine the melting temperature (Tm) for both 3WJ and 4WJ and ensure that the nanostructures will not unfold at physiological temperature (37 °C). The core structure of both 3WJ and 4WJ was designed to have a Tm above 72 °C, and the Tms for 3WJ and 4WJ were found to be 79 and 77 °C, respectively (Figure 2c). Using DLS, we measured the diameter of bare 3WJ and 4WJ to 12.68 and 13.45 nm, respectively, which is close to the estimated diameter of 16.32 nm for both the 3WJ and 4WJ core (Appendix A). There is no significant difference between the diameter of the bare 3WJ and 4WJ. 

We expected the measured size of 3WJ and 4WJ from DLS to be similar since the instrument measures the hydrodynamic radius, and we assumed that the nanostructures are spherical and the arm lengths of the two structures are identical. 

#### 3.1.3. Cargo Loading and Characterization 

The fluorescence characterization was carried out with bare 4WJ, 4WJ with fluorophore (4WJ-1F), and 4WJ with both fluorophore and quencher (4WJ-1FQ) to ensure the 4WJ-1FQ had minimum fluorescence compared to the 4WJ-1F. The quenchers were able to reduce the fluorescence intensity of HEX, FAM, and Texas Red (TR) by 90%, 93%, and 98%, respectively. The fluorescence intensity curves for bare 4WJ and 4WJ-HEX with or without quencher are shown in Figure 2d, and the fluorescence intensity curves for all other constructs (with different fluorophores and with or without quencher) can be found in Appendix A. Additionally, the fluorescence intensity curve validating the modification of the 3WJ system can also be found in Appendix A.

### 3.2. Quantifing DBCO Linker Modification of Pure Gelatin and DNA Bait Modification of DG

Gelatin was first modified with a DBCO linker (DBCO-PEG4-NHS ester) via NHS/amine coupling. DBCO-PEG4-NHS ester was selected because the NHS ester is reactive towards the amine residuals presented on gelatin to incorporate DBCO linker, which can be used for copper-free click chemistry [49]. The gelatin was successfully modified with the DBCO-PEG4-NHS ester and the degree of substitution of DBCO-gelatin (DG) was calculated using a fluorescamine assay. 

This assay was used to quantify the number of reactive amine group; thus, a decrease in fluorescamine fluorescence would indicate conjugation on the gelatin amine groups. A representative fluorescamine fluorescence curve for the glycine standard and gelatin samples can be found in Appendix A. The amount of DBCO linker presented can be calculated based on the difference of reactive amine groups between modified and unmodified gelatin. Based on the fluorescamine assay result, we were able to modify 116.38 ± 35.80 µM of reactive amine groups using 135 µM of DBCO-PEG4-NHS ester, which converts to about an 86% reaction efficiency. The values for each measurement for DBCO linker quantification can be found in Appendix A.

Azide-modified DNA bait was attached to DG via copper-free click chemistry, which was chosen due to its biocompatibility, bioorthogonality, and high efficiency [50,51,52]. The conjugation of DNA baits allows for the attachment of DNA motifs to the gelatin polymer. The DNA bait attachment was quantified with fluorescence measurement after hybridizing a complementary DNA strand with HEX fluorophore (bait probe). The DNA sequence for both DNA bait and probe can be found in Appendix A. 

The mixture of bait modified gelatin and bait probe was incubated in a thermocycler from 37 to 4 °C over 25 min to facilitate DNA hybridization, and any unreacted DNA probe was filtered out afterward. A sample fluorescence curve for the quantification can be found in Appendix A. Based on the fluorophore results, we were able to modify 27.04 ± 9.97 µM of the DBCO with 45 µM of DNA bait, which converts to a 60.08% reaction efficiency for the DNA bait attachment. The values for each measurement for DNA bait quantification can be found in Appendix A.

### 3.3. Calibration and Development of DNA Motif for Strand Displacement Release Circuit

#### 3.3.1. Calibration of Strand Displacement Release Circuit in Solution 

The main design goal for the DNA motif is the ability to trigger the release of specific cargo based on the input trigger, and each trigger is exclusive for the specific cargo; therefore, each trigger should only allow for the release of one specific cargo. The specificity of the release triggers was verified by monitoring the change in fluorescence intensity with the various trigger/4WJ combinations over time. Each 4WJ was assembled with one fluorophore (HEX, FAM, or TR) and the respective quencher attached to it. 

With the addition of the corresponding trigger solution, a rapid increase in fluorescence was observed, and no increase in fluorescence for the non-complementary triggers were observed (Appendix A). This corresponds to other studies that showed the highly specific nature of DNA strand displacement and met our design specifications [41,43,53]. Additionally, the release curve was fitted with a one phase association model to better model the release behavior. The 4WJ-HEX release curve in solution and the fitted model are shown in Figure 3a. 

Additional release curves and fitted models for 4WJ-FAM and 4WJ-TR can be found in Appendix A. The 50% release time (T_50_) on average was 10.6 min for the 4WJ release in solution. All T_50_ measured for each complex can be found in Appendix A. It is important to note that release in our system was slower than some previous studies observed [40,41]; however, the salt condition was also different, which might impact the hybridization process of DNA and likely cause the change observed in the release profile.

In addition to the release behavior for trigger strands that perfectly match their complementary target strand, we designed mismatched trigger strands that could allow for further modulation of the release behavior. Furthermore, these experiments were used to evaluate how base mismatches in strand displacement can affect the release behavior of cargos and help to determine design rules that could inform future trigger design to precisely control the release time and the amount released. We designed two categories of mismatch triggers based on the location of the mismatches: a 3′-end mismatch trigger and center mismatch trigger. 

A schematic of the design is shown in Figure 3b. The DNA sequence for the mismatch triggers can be found in Appendix A, and the mismatched trigger release curves for the center mismatch triggers and 3′ mismatched trigger are shown in Figure 3c,e, respectively. There are observable differences of the final release quantities for both categories of mismatch triggers; however, only the center mismatch trigger group has an observable difference in the rate of release. Additionally, with increasing the number of mismatches within the trigger, there is a significant decrease in the final release percent for both center and 3′ mismatched triggers. 

The full Dunn’s Kruskal–Wallis multiple comparison post-hoc analysis table can be found in Appendix A. The graphs for final release for the center and 3′ mismatched triggers can be found in Figure 3d,f, respectively. However, there was no significant decrease in release observed between the regular trigger with either center or 3′ end mismatch trigger with one mismatched base. This is likely because the other base pairings can overcome the single mismatch and continue the strand displacement process; nonetheless, it is noticeable that the release rate was slower for the center mismatch trigger with one mismatched base even though there was no significant difference in the final release quantity.

#### 3.3.2. Developing the Release Circuit in Hydrogel 

After the calibration and characterization study within the solution, 4WJ was incubated with DNA bait-modified gelatin, which allows the designated overhang on 4WJ to hybridize with the DNA bait, to obtain DNA-Gel. Prior to the formation of DNA-Gel hydrogel, we validated the EDC/NHS crosslinking method, confirmed the hydrogel formation with pure gelatin hydrogel and determined a swelling ratio of 123 ± 5%. An image of both pure gelatin and DNA-Gel hydrogel and the bar graph for the swelling ratio study can be found in Appendix A. 

The release behavior of 4WJ-HEX within the gelatin hydrogel was studied by mixing the DNA-Gel with pure gelatin and crosslinked with EDC/NHS to form a DNA-Gel hydrogel. The fluorescence was measured every 30 min over the course of 17 h. The release curve for 4WJ-HEX and the fit for the release are shown in Figure 4a along with an image of the hydrogel. The T_50_ for 4WJ-HEX is approximately 2 h, which is much longer than the T_50_ observed in-solution. 

The increase in release time is likely due to the diffusion limitations of the system since the trigger will need to diffuse into the DNA-Gel hydrogel for the strand displacement reaction to occur. The release rate of our system is notably faster than some of the other release systems reported in other studies, which typically have a release time over a few days [54,55,56,57]. However, it is important to note that most of these studies characterized the release behaviors of large biomolecules, such as erythropoietin (~30.4 kDa) and vascular endothelial growth factor (~19–22 kDa), which has a slower diffusive rate within a hydrogel. Additionally, another consideration is the crosslink density, which affects the pore size of the hydrogel and the diffusive rate of the molecules within, which might contribute to the slower release reported in previous studies. 

Further examination of the hydrogel properties will be required to untangle the effect of the crosslink density of our DNA-Gel hydrogel system on the release. Furthermore, it should be noted that, within our system, the release is measured by the increase in fluorescence within the hydrogel and not the supernatant, since the quencher was released instead of the fluorophores. Therefore, the actual release from the gel would be slower because the released strand would need to previously diffuse inside the hydrogel to reach its target. Representative fluorescent emission curves for the 4WJ-HEX release in DNA-Gel hydrogel can be found in the Appendix A. 

In addition to the single release behavior in the gelatin hydrogel environment, an on-demand sequential release behavior was demonstrated using 4WJ-2FQ with FAM and TR. The fluorescent curves for both FAM and TR were acquired every 30 min in the study. The release of specific cargo is clearly shown in the release curve where the first trigger addition caused the increase of fluorescence in the FAM channel and no change in fluorescence of TR until the second trigger was added after eight hours, which is shown in Figure 4b. 

The T_50_ for both FAM and TR is approximated to 2 h. It is notable that the T_50_ for all three fluorophores is similar, which is likely because all the triggers used have a similar molecular weight (~9.9 kDa) and therefore should exhibit a similar diffusion rate within the DNA-Gel hydrogel system. The dip in the release curve for FAM at the eighth hour can be explained by the experimental setup that required us to move the plate out of the instrument to add the second trigger before resuming the measurements. 

It is important to note that other sequential release mechanisms have been reported in the literature based on different stimuli and physical properties [58,59]. However, those mechanisms tend to require specific peptide synthesis/modification or specific differences in the physical properties of the molecules, such as the hydrophobicity. Our strategy could potentially overcome those limitations by utilizing a specific aptamer for biomolecule incorporation and release via a strand displacement reaction. 

Moreover, to ensure that the release observed in the DNA-Gel hydrogel is not due to the degradation of 4WJ and validate the applicability of our strategy for future biological applications, the stability of 4WJ was assessed in physiological conditions in the presence of serum. DNA-Gel hydrogel with a quencher was assembled, and either mouse serum (20% in PBS) or DNAse I was added on top of the hydrogel. 4WJ-HEX in solution with DNAse was used for comparison. As shown in Figure 4c, the degradation of 4WJ-HEX in solution was faster than the degradation of 4WJ-HEX within a DNA-Gel hydrogel. 

After 24 h, the degradation of the 4WJ embedded within the DNA-Gel hydrogel is only ~20% in the serum condition (Figure 4c), while complete degradation in solution occurs around 4–24 h for bare DNA nanostructures [60,61,62]. This is likely because the DNA nanostructure is embedded within the hydrogel, and the nucleases within the serum need to diffuse through the hydrogel to start the degradation process. The nanostructures might also be protected by the environment created by the DNA-Gel hydrogel. This indicates that our DNA-Gel hydrogel system should remain stable for a few days, which means that it could be useful for several cell-based assays. Additionally, this indicates that the release we observed during the controlled release study was not caused by the degradation of the 4WJ structures but from the strand displacement reaction. Representative fluorescence curves for the stability study can be found in the Appendix A.

## 4. Conclusions

A chemically crosslinked gelatin hydrogel with incorporated DNA motifs (DNA-Gel) was successfully fabricated using EDC/NHS chemistry for crosslinking and copper-free click chemistry for DNA nanostructure attachment. Additionally, the highly specific and on-demand release of payloads using DNA nanotechnology, namely the DNA nanostructure and strand displacement reaction, was demonstrated in the hydrogel. In addition, our results showed that the release rate was much slower in the hydrogel in comparison with the release observed in solution, which is likely due to the slower diffusion rate of the trigger within the hydrogel. 

Modulation of the release behavior was also achieved by the incorporation of mismatched bases within the trigger strand, which can potentially be incorporated into future design to further tune the release behavior. Furthermore, the ability for multiplex release using a strand displacement circuit was demonstrated within the hydrogel system. An amplification circuit can be incorporated to further modulate the release behavior. A DNA logic system can also be implemented to generate a more complex release system. 

Overall, this strand displacement circuit inserted in hydrogel with multiplexing potential could be used to release specific biomolecules (e.g., cytokines and growth factors) in a specific cascade to control cell behaviors, which would improve the capabilities of the current scaffold for tissue engineering. In addition, this system is readily applicable to other biopolymer-based hydrogels, such as collagen, and could be used in bioink formulations.

## Figures and Tables

**Figure 1 biomedicines-10-00213-f001:**
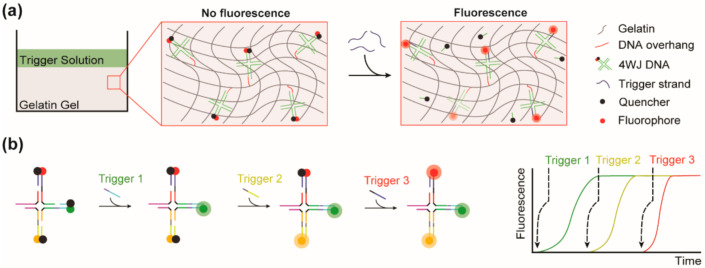
Graphical representation of the DNA-Gel hydrogel system and the SDR-based sequential release mechanism. (**a**) Schematic showing the DNA motifs inserted in the gelatin hydrogel via click chemistry and illustrating the release mechanism within the gelatin hydrogel using strand displacement reaction. (**b**) Schematic of the sequential release mechanism developed on the DNA motifs showing specific release of the fluorescent cargo based on the sequential use of trigger strands. The increase in fluorescence indicates the separation of the quencher and the fluorophore. The quencher and fluorophore are used to quantify the cargo release in our system.

**Figure 2 biomedicines-10-00213-f002:**
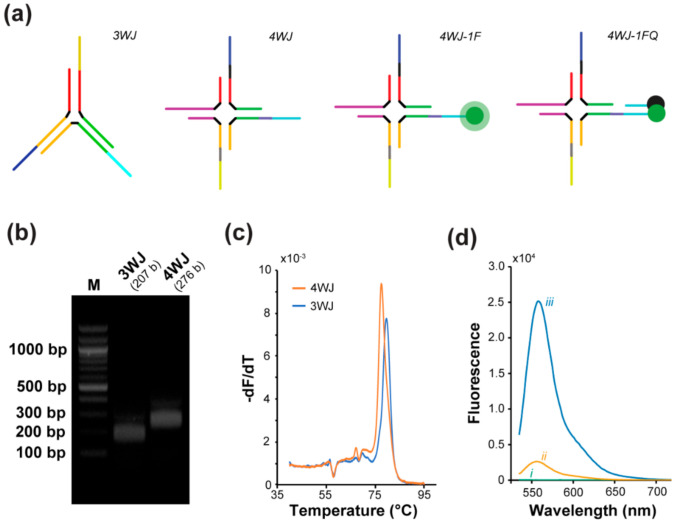
DNA nanostructures characterization. (**a**) Schematic of the bare 3WJ, 4WJ, 4WJ with one fluorophore (4WJ-1F), and 4WJ with one fluorophore and one quencher (4WJ-1FQ) design. (**b**) 1.5 wt% agarose gel electrophoresis for both bare 3WJ and 4WJ. (**c**) The melting curve for both bare 3WJ and 4WJ. (**d**) Fluorescence characterization of the different 4WJ states. (i) 4WJ, (ii) 4WJ-1FQ, and (iii) 4WJ-1F.

**Figure 3 biomedicines-10-00213-f003:**
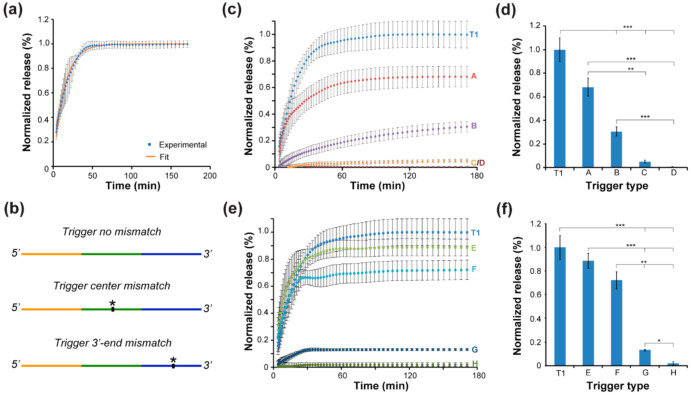
4WJ controlled release in solution (**a**) 4WJ-HEX release curve with regular trigger in solution. (**b**) Schematic of the mismatch design. The mismatch location is denoted with *. (**c**) 4WJ-HEX release curve with varying center mismatched trigger. (**d**) Bar graph representation of the release curve shown in (**b**) (Dunn’s Kruskal–Wallis multiple comparison, ** = *p* < 0.01, *** = *p* < 0.001). The trigger type label corresponds to the label shown in Appendix A. (**e**) 4WJ-HEX release curve with varying 3′ mismatched trigger. (**f**) Bar graph representation of the release curve shown in (**b**) (Dunn’s Kruskal–Wallis multiple comparison, * = *p* < 0.05, ** = *p* < 0.01, and *** = *p* < 0.001). The labeling of trigger type is corresponding to the DNA sequence shown in Appendix A. The release was normalized against the positive control, which is the DNA motif without the quencher.

**Figure 4 biomedicines-10-00213-f004:**
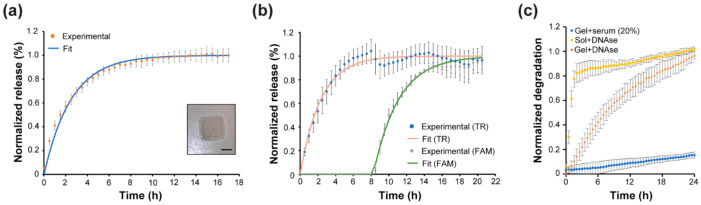
Controlled release within DNA-Gel hydrogel. (**a**) 4WJ-HEX release curve within DNA-Gel hydrogel. Inset shows the crosslinked gelatin hydrogel. Scale bar: 1 mm. (**b**) The sequential release of 4WJ-2FQ within in DNA-Gel hydrogel. (**c**) Stability of the 4WJ-HEX against serum in DNA-Gel hydrogel and DNase within solution and DNA-Gel hydrogel. The release was normalized against the positive control, which is the DNA motif without the quencher.

**Table 1 biomedicines-10-00213-t001:** Component table for both the pure gelatin hydrogel and DNA-Gel hydrogel. The volume represents the amount needed for 100 µL of hydrogel, and the initial concentration of each component is shown in parentheses. N/A indicates that the component was not used in the hydrogel.

Hydrogel Type	Gelatin Solution (10% *w*/*v*)	DNA-Gel Solution (0.01% *w*/*v*)	EDC/NHS (962.5 mM)	1× PBS
Pure Gelatin Hydrogel	50 µL	N/A	2 µL	48 µL
DNA-Gel Hydrogel	50 µL	20 µL	2 µL	28 µL

## Data Availability

All data generated in this study will be available upon reasonable request to the corresponding author.

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
