# Peer review of "Controlled Release in Hydrogels Using DNA Nanotechnology"

_biomedicines, 2022, doi:10.3390/biomedicines10020213_

Round 1

Reviewer 1 Report

In this article, the authors described a method to design controlled release of different substances in a hydrogel system based on DNA nanotechnology. The work has been written nicely and the results are also well presented. Here, I only had a few comments for the manuscript before its acceptance.

  1. The authors used the normalized release to present the controlled release of different chemicals. However, the definition of the normalized values should be clarified.
  2. Reference errors on lines 237, 254, 273, etc.
  3. Formatting problem. Figure 3 appeared three times in the manuscript.

Reviewer 2 Report

The paper entlited “Controlled release in hydrogels using DNA nanotechnology” studies a DNA-based circuit for temporal release of biomolecules that can be integrated in hydrogels.

The work is very difficult in understanding and confusing in reading both in the experimental and in the results section.

Some remarks:

  1. 2.7 Gelatin and 4WJ-Gelatin (DNA-Gel) Hydrogel Formation

The method of preparation should be better explained, reference is made to a 2007 paper (bibliographic reference n.44) in which DNA is not used but pilocarpine. A table with the different concentrations of all hydrogel components is missing.

  1. the mechanical and technological characterization of the hydrogel is missing.
  2. Figure 1 is not well explained
  3. the text is not cured in form, there are many errors such as:

Line 237 Error! Reference source not found.a,

line 254  Error! Reference source not found.b

line 273  Error! Reference source not found.c

lines 325, 343, 352, 360 is it always the same figure 3 repeated 4 times?

Tables and figures not present in the text but present in Supplementary Materials are cited.

Reviewer 3 Report

The work entitled “Controlled release in hydrogels using DNA nanotechnology” by Hu an Veneziano introduces a new designed and synthesized versatile multi-arm DNA motifs that can be easily conjugated within a gelatin hydrogel via click chemistry to incorporate a strand displacement circuit. The work is very well introduced, and the methodology employed extremely detailed. The results are also clear, even though it sometimes makes it difficult to concentrate on the information considering there is so much data in the supporting information file. Regardless, the work has merit and is scientifically sound. The authors did a very good job and the quality and subject of this work fits that of the journal, and for that this reviewer recommends this manuscript publication after minor revision.

In detail:

- the introductory portion of the abstract is too extensive, leaving little space for the actual major findings of the work. These should be better emphasized

-the novelty of the research should be highlighted in the introduction

-there is a problem with the references that should be fixed

- figure 3 appears 4 times, making it a bit difficult to understand the information. There is no (c) in the image.

- there are too many tables and figures in the supporting information file, making the constant consultation of the data a bit difficult to keep the attention on the manuscript. The authors could put some of the data on the manuscript to facilitate reading.

- English writing must be improved since writing mistakes were detected.

Round 2

Reviewer 2 Report

Dear Authors,

I appreciate the changes, corrections and additions that have been made to the manuscript.